# Laser Requirements for High-Order Harmonic Generation by Relativistic Plasma Singularities †

**Alexander S. Pirozhkov** [1,*], **Timur Zh. Esirkepov** [1], **Tatiana A. Pikuz** [2,3],
**Anatoly Ya. Faenov** [3,4], **Akito Sagisaka** [1], **Koichi Ogura** [1], **Yukio Hayashi** [1], **Hideyuki Kotaki** [1],
**Eugene N. Ragozin** [5,6], **David Neely** [7,8], **James K. Koga** [1], **Yuji Fukuda** [1], **Masaharu Nishikino** [1],
**Takashi Imazono** [1], **Noboru Hasegawa** [1], **Tetsuya Kawachi** [1], **Hiroyuki Daido** [9], **Yoshiaki Kato** [10],
**Sergei V. Bulanov** [1,6,11,12], **Kiminori Kondo** [1], **Hiromitsu Kiriyama** [1] and **Masaki Kando** [1]

1   Kansai Photon Science Institute (KPSI), National Institutes for Quantum and Radiological Science and Technology (QST), Kizugawa, Kyoto 619-0215, Japan; Timur.Esirkepov@qst.go.jp (T.Zh.E.); sagisaka.akito@qst.go.jp (A.S.); ogura.koichi@qst.go.jp (K.O.); hayashi.yukio@qst.go.jp (Y. H.); kotaki.hideyuki@qst.go.jp (H.Ko.); koga.james@qst.go.jp (J.K.K.); fukuda.yuji@qst.go.jp (Y.F.); nishikino.masaharu@qst.go.jp (M.N.); imazono.takashi@qst.go.jp (T.I.); hasegawa.noboru@qst.go.jp (N.H.); kawachi.tetsuya@qst.go.jp (T.K.); Sergei.Bulanov@eli-beams.eu (S.V.B.); kondo.kiminori@qst.go.jp (K.K.); kiriyama.hiromitsu@qst.go.jp (H.Ki.); kando.masaki@qst.go.jp (M.K.)
2   Graduate School of Engineering, Osaka University, Suita, Osaka 565-0871, Japan; pikuz.tatiana@gmail.com
3   Joint Institute for High Temperatures of the Russian Academy of Sciences, 125412 Moscow, Russia; anatolyf@hotmail.com
4   Open and Transdisciplinary Research Initiatives, Osaka University, Suita, Osaka 565-0871, Japan
5   P. N. Lebedev Physical Institute of the Russian Academy of Sciences, 119991 Moscow, Russia; enragozin@gmail.com
6   Moscow Institute of Physics and Technology (State University), 9 Institutskii pereulok, Dolgoprudnyi, 141700 Moscow Region, Russia
7   Central Laser Facility, Rutherford Appleton Laboratory, STFC, Oxon OX11 0QX, UK; david.neely@stfc.ac.uk
8   Department of Physics, SUPA, University of Strathclyde, Glasgow G4 0NG, UK
9   Naraha Remote Technology Development Center, JAEA, Yamadaoka, Naraha, Futaba, Fukushima 979-0513, Japan; daido.hiroyuki@jaea.go.jp
10  The Graduate School for the Creation of New Photonics Industries, Nishiku, Hamamatsu, Shizuoka 431-1202, Japan; y.kato@gpi.ac.jp
11  A. M. Prokhorov Institute of General Physics of the Russian Academy of Sciences, 119991 Moscow, Russia
12  Institute of Physics of the Czech Academy of Sciences, v.v.i. (FZU), ELI-Beamlines project, 182 21 Prague, Czech Republic
*   Correspondence: pirozhkov.alexander@qst.go.jp; Tel.: +81-774-71-3371
†   This paper is an extended version of our paper to be published in the 15th International Conference on X-Ray Lasers (ICXRL2016), Nara, Japan, 2016.

**Abstract:** We discuss requirements on relativistic-irradiance ($I_0 > 10^{18}$ W/cm$^2$) high-power (multi-terawatt) ultrashort (femtosecond) lasers for efficient generation of high-order harmonics in gas jet targets in a new regime discovered recently (Pirozhkov et al., 2012). Here, we present the results of several experimental campaigns performed with different irradiances, analyse the obtained results and derive the required laser parameters. In particular, we found that the root mean square (RMS) wavefront error should be smaller than ~100 nm (~$\lambda/8$). Further, the angular dispersion should be kept considerably smaller than the diffraction divergence, i.e., µrad level for 100–300-mm beam diameters. The corresponding angular chirp should not exceed $10^{-2}$ µrad/nm for a 40-nm bandwidth. We show the status of the J-KAREN-P laser (Kiriyama et al., 2015; Pirozhkov et al., 2017) and report on the progress towards satisfying these requirements.

**Keywords:** high-power femtosecond lasers; high-power laser quality; relativistic laser plasma; relativistic plasma singularities; coherent X-ray generation; burst intensification by singularity-emitting radiation

---

## 1. Introduction

In the new regime of coherent X-ray generation [1,2], intense ($>10^{18}$ W/cm$^2$) high-power (multi-TW) femtosecond laser pulses focused onto gas targets propagate through underdense plasma (electron density $n_e$ ~$10^{19}$ cm$^{-3}$) and produce an electron-free cavity and bow wave [3]. The resulting multi-stream relativistic plasma flow leads to the formation of density singularities (cusp catastrophes) located at the joining of the cavity wall and the bow wave front. The robustness and structural stability of these singularities are explained with catastrophe theory [4,5]. These singularities, oscillating under the action of an intense laser field, emit coherent high-frequency (up to the soft X-ray spectral region) radiation via the Burst Intensification by Singularity-Emitting Radiation (BISER) mechanism [6]. This emission originates from extremely localized, nanometre-scale regions where the local electron density is several orders of magnitude higher than the original plasma density. This results in constructive interference, i.e., X-ray yield proportional to the number of electrons squared. The large number of photons together with the small source size and attosecond duration provide very high brightness, e.g., $10^{27}$ photons/mm$^2$·mrad$^2$·s in 0.1% bandwidth at a wavelength of 18 nm [6].

Here, we report on the results of several experimental campaigns with the J-KAREN laser [7], which demonstrated strong dependence of the X-ray yield on the irradiance. We analyse the obtained dependence and derive laser parameters required for efficient coherent X-ray generation in this regime. As high-power lasers typically suffer from wavefront distortions and spatiotemporal couplings [8–10], which decrease the irradiance, all these imperfections should be kept at a minimum. In particular, for noise-like wavefront distortions, the root mean square (RMS) wavefront error should be well below 100 nm. The angular dispersion, i.e., the dependence of propagation direction on wavelength, should be minimized, so that the difference between propagation angles could be kept significantly smaller than the diffraction divergence, i.e., μrad level for 100–300-mm beam diameters. The corresponding angular chirp, i.e., the derivative of the propagation angle on wavelength, should not exceed ~$10^{-2}$ μrad/nm for a 40-nm bandwidth. We show the status of the J-KAREN-P laser [11] and report on the progress towards satisfying these requirements [12].

## 2. Materials and Methods

We performed several experimental campaigns with the J-KAREN laser [7], where the experimental setup and the detection system were approximately the same. Figure 1 shows typical experiment scheme. The laser wavelength was ~0.8 μm; the pulse energy varied from 0.4 to 0.8 J; the effective pulse duration from 34 to 160 fs; and the peak power from 5 to 20 TW. The pulses were focused with an $f/9$ off-axis parabolic mirror onto a supersonic helium gas jet; typical gas jet density profiles are shown in Figure 2 of [2]. The estimated peak irradiance (in vacuum) varied due to the laser pulse power and focal spot size difference from ~$2 \times 10^{18}$ W/cm$^2$ to ~$10^{19}$ W/cm$^2$. Figure 2 shows the focal spot shapes corresponding to the lowest and highest irradiances. All the data described here were obtained with linearly-polarized pulses, which generate X-rays via the BISER mechanism more efficiently than the circularly-polarized ones [2].

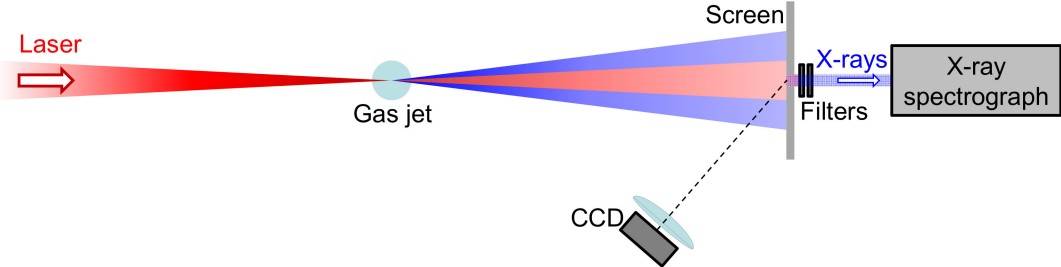

**Figure 1.** Typical experiment scheme. The laser pulse irradiates the He gas jet target and generates coherent soft X-rays detected with the X-ray spectrograph. In some experimental campaigns, the transmitted laser beam footprint was measured with the Teflon screen and optical charge-coupled device (CCD); the X-rays passed to the spectrograph through the aperture made in the screen.

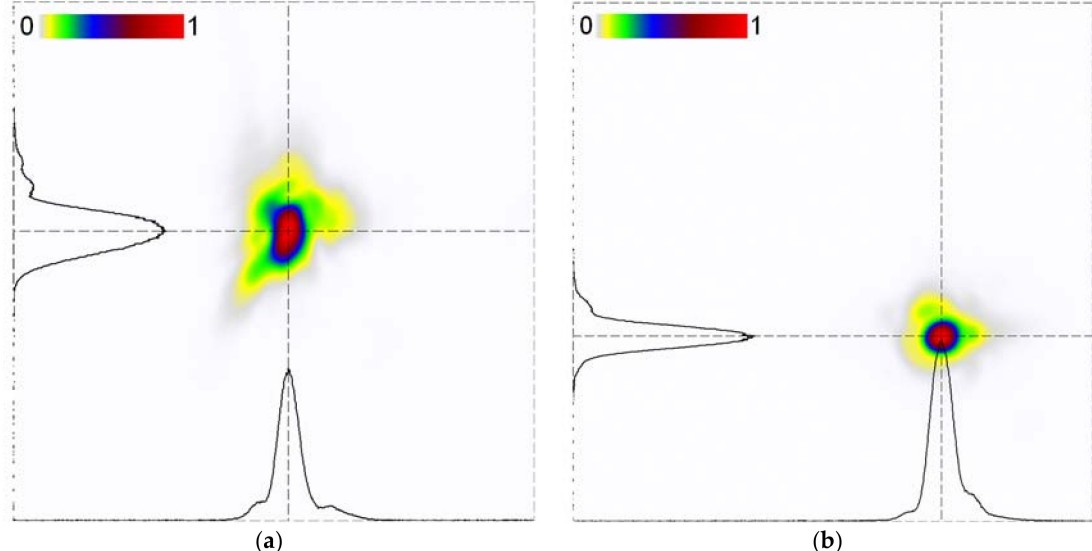

**Figure 2.** Focal spot shapes corresponding to (**a**) the largest and (**b**) smallest spots used in the experiments. The effective spot radii are 10.1 and 6.9 μm, while the Full Width at Half Maximum (FWHM) sizes are 9.4 μm × 17.6 μm and 10.0 μm × 9.4 μm, respectively. The full frame size is 200 μm, and the laser polarization is horizontal for both (**a**,**b**).

The X-ray spectra were recorded in all the experiments in the forward direction with a flat-field grazing-incidence spectrograph comprised of a toroidal mirror, slit, optical blocking filters [13], flat-field varied-line-space spherical grating [14] and back-illuminated charge-coupled device (CCD). The spectrograph is described in more detail in [2].

Harmonic generation by the BISER mechanism depended on several factors, most importantly on the plasma density and gas jet position with respect to the laser focus. The details of these dependences will be reported elsewhere. In each experimental campaign, the harmonic generation conditions were optimized varying the plasma density and gas jet position. In a wide region of parameter space around these optimum conditions, the BISER was a dominant mechanism of X-ray generation in the spectral region of interest. We note that our plasma length was kept relatively short, typically ~1–2 mm, which was significantly shorter than in experiments for electron acceleration and betatron X-ray generation with similar lasers [15].

## 3. Results

### 3.1. Dependence on the Irradiance

The strongest spectra obtained in each experimental campaign are shown in Figure 3. Curve (a) corresponds to the experimental campaign performed with the focal spot shown in Figure 2a (the lowest irradiance among all the campaigns), while Curve (b) corresponds to the campaign performed at the highest irradiance achieved with the focal spot shown in Figure 2b. Curve (1) is the spectrum shown in Figure 2d of [1]; this spectrum was obtained with the irradiance of ~ $4 \times 10^{18}$ W/cm$^2$. The dips in the spectra (1) and (b) at 284 eV were due to the enhanced absorption of optics contamination at the C absorption edge; this edge is the beginning of the 'water window' spectral range. Despite similar laser parameters, i.e., with the ratio of the maximum and minimum pulse energies, powers and irradiances of just several times, the achieved X-ray yields were drastically different, by orders of magnitude. The difference is especially important at higher photon energies, e.g., in the 'water window' spectral range, where the spectrum (a) is below the detection threshold and thus cannot be used for experiments and applications with similar-sensitivity spectrographs; we note here that our spectrograph was rather sensitive, and making a spectrograph with several orders of magnitude higher sensitivity, which would be sufficient to record the 'water window' part of the spectrum for the Curve (a), would be a challenging task.

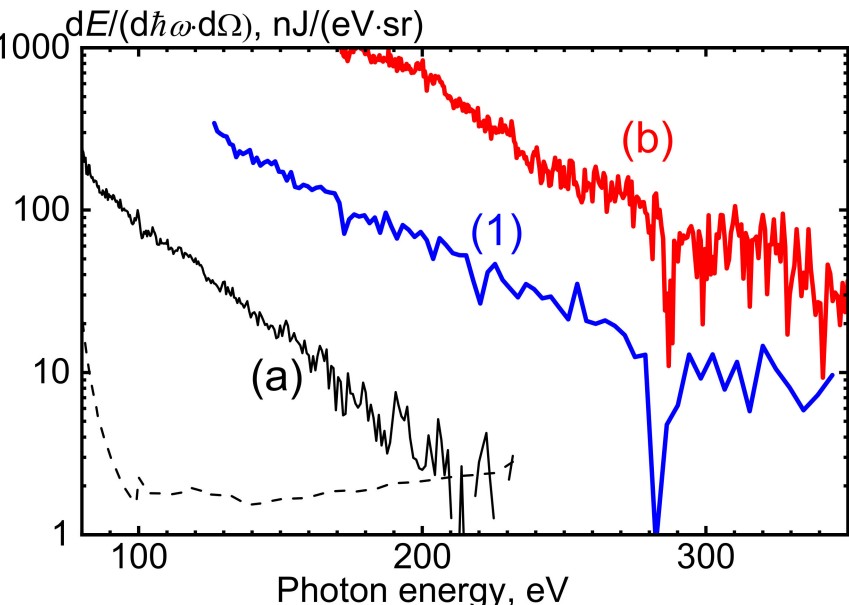

**Figure 3.** The strongest X-ray spectra obtained in each of the experimental campaigns performed with the same laser (J-KAREN). Curves (a) and (b) are obtained in the experimental campaigns with the spots shown in Figure 2a,b, respectively. Curve (1) is the spectrum shown in Figure 2d of [1]. The dashed curve is the estimated noise level for Curve (a).

### 3.2. Driving Laser Requirements

The steep growth of the X-ray yield with the irradiance looks rather promising; it is of interest to see whether the harmonic yield continues to grow rapidly at irradiances higher than $10^{19}$ W/cm$^2$. However, for a given laser power and experimental setup, the irradiance is always limited. The highest irradiance is achieved when the focal spot approaches the diffraction limit. In practice, many effects reduce the irradiance. Here, we discuss the requirements on the wavefront and angular chirp, which are two imperfections that are very typical for high-power lasers.

For noise-like high-frequency wavefront distortions, the irradiance can be estimated as:

$$I = I_0 \exp\left[-\left(\frac{\eta\pi\sigma}{\lambda}\right)^2\right],\qquad(1)$$

where $I_0$ is the diffraction-limited irradiance, $\sigma$ is the wavefront distortion RMS, $\lambda$ is the laser wavelength and the factor $\eta \approx 2$ [16]. We found that for our wavefront sampling and wavefront distortion statistics, the value $\eta \approx 1.85$ gave a good fit to the data; see Figure 4, where we show the normalized irradiance, $I/I_0$, obtained from the Point-Spread Function (PSF) calculated from the measured wavefront.

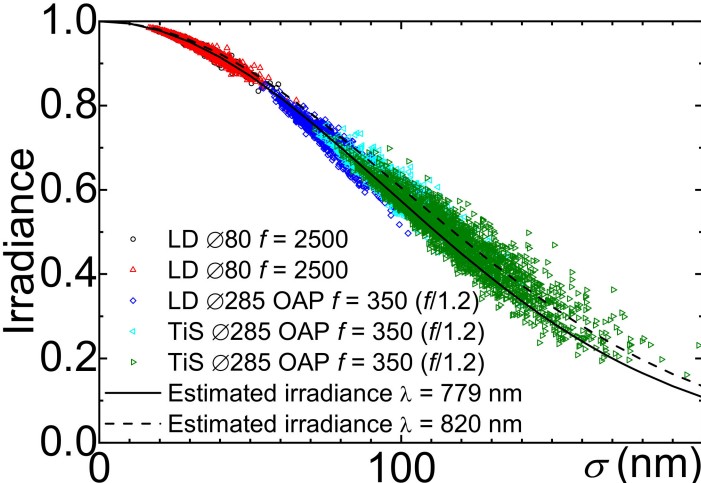

**Figure 4.** Dependencies of the normalized irradiance calculated from the measured wavefront on the root mean square (RMS) wavefront distortion $\sigma$. The data are obtained with two lasers: Laser Diode (LD), $\lambda$ = 779 nm and Ti:sapphire (TiS), $\lambda$ = 820 nm, beam diameters of 80 and 285 mm, focused with an $f$ = 2500 mm lens and $f$ = 350 mm ($f$/1.2) Off-Axis Parabolic (OAP) mirror. The solid and dashed lines show the estimates using (1) with $\eta$ = 1.85.

Comparing the dependence of the normalized irradiance on the wavefront error with the dependence of the X-ray yield on the irradiance, we conclude that for efficient harmonic generation, the wavefront distortion should be kept smaller than about 100 nm RMS, i.e., smaller than $\lambda/8$, which is close to the Maréchal criterion of a diffraction-limited beam. Lower-order aberrations should be minimized, as well.

Angular dispersion, i.e., the dependence of the propagation direction on wavelength, is a kind of spatiotemporal coupling [8–10], which distorts the focal spot and elongates the pulse duration. Most commonly, the angular dispersion is caused by the stretcher or vacuum compressor misalignment or by passing through wedged optics. To keep the spot nearly diffraction-limited, the divergence caused by angular dispersion should be well below the diffraction divergence. For linear angular chirp $C$ (µrad/nm), this gives the requirement:

$$C\Delta\lambda \ll \frac{\lambda}{D},\qquad(2)$$

where $\Delta\lambda$ is the bandwidth and $D$ is the beam diameter. Thus, this requirement is particularly severe for high-power femtosecond lasers, which have simultaneously large bandwidth and beam diameter. For the J-KAREN-P laser (final beam diameter of ~280 mm, bandwidth of ~40 nm), this gives $C \ll 0.075$ µrad/nm. For example, to keep irradiance >90% of the ideal case, there should be $C < 10^{-2}$ µrad/nm; see Figure 5a. In order to satisfy this requirement, all angles of all compressor gratings should be aligned with the accuracy of the order of 10 µrad. For example, the dependence of

the irradiance on the groove tilt, i.e., rotation of the grating around the normal to its surface, is shown in Figure 5b. Similar dependences exist for the other two angles, i.e., the grating rotation around the groove direction and rotation around the in-plane axis perpendicular to the grooves. We note that the irradiance estimates shown in Figure 5 include both the effect of spot increase and pulse elongation.

We also emphasize that the sensitivity of the harmonic yield to the irradiance means that the shot-to-shot fluctuations of the irradiance should be minimized.

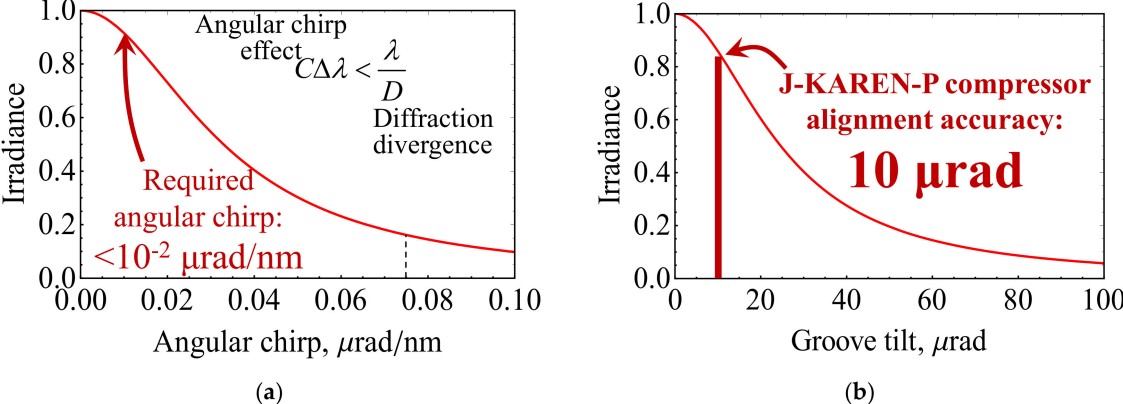

(a)    (b)

**Figure 5.** Influence of the angular dispersion and compressor misalignment on the irradiance. These examples are calculated for the J-KAREN-P compressor ($\lambda$ = 805 nm, $\Delta\lambda$ = 38 nm, 284 mm × 256 mm beam, 1480 lines/mm gratings at the incidence angle $\alpha$ = 49.6°); the estimates are based on [8]. (**a**) Estimated irradiance vs. amount of angular chirp; the dashed line corresponds to $C_{DL} = \lambda/(\Delta\lambda \cdot D) \approx 0.075$ µrad/nm. (**b**) Estimated irradiance vs. compressor misalignment; here, the case of a groove tilt of one grating.

### 3.3. J-KAREN-P Laser Preparation

We have recently upgraded the J-KAREN-P laser, which can now supply petawatt pulses at a repetition rate of 0.1 Hz [11]. To ensure efficient coherent X-ray generation by the BISER mechanism, we are taking measures to achieve good laser parameter stability and high spot quality. In particular, we set up a deformable mirror after the final amplifier, at the 90 mm beam diameter stage. We also aligned all 12 angles of the four compressor gratings with an accuracy of ~10 µrad, as required for attaining high irradiance; see Figure 5. As a result, we achieved the wavefront error of ~70 nm RMS (280-mm diameter J-KAREN-P beam before the compressor) and a focal spot approaching the diffraction limit, i.e., ~50% of the diffraction-limited irradiance, measured at 300 TW power. We also achieved pulse compression down to the bandwidth limit, i.e., ~90% of the bandwidth-limited power. These results are described in detail in [12]. Here, we report on the beamline stabilization and discuss in more detail the difference between the real focal spot (as measured in [12]) and the one calculated from the wavefront.

Figure 6 shows the on-target energy stability in the power amplifier mode; the data are of 558 shots taken during four hours of operation. The pulse energy fluctuation was 1.8% RMS. We note that readout fluctuations of a typical energy sensor are larger than or comparable to this value. Further, the energy measured via the calibrated near-field camera suffers from calibration uncertainty due to the spectral shape variations, as the filters' transmission and camera sensitivity depend on the wavelengths. To avoid the influences of these problems, we used a fibre spectrometer measuring the zero-order reflection from the compressor grating; the full beam was collected and sent to an integrating sphere connected to the spectrometer by an optical fibre. The wavelength dependence of the sensitivity of the whole setup was calibrated using a black body light source, and the absolute calibration was performed using a comparison with an energy sensor in the averaging mode, effectively reducing the effect of fluctuations on the calibration accuracy.

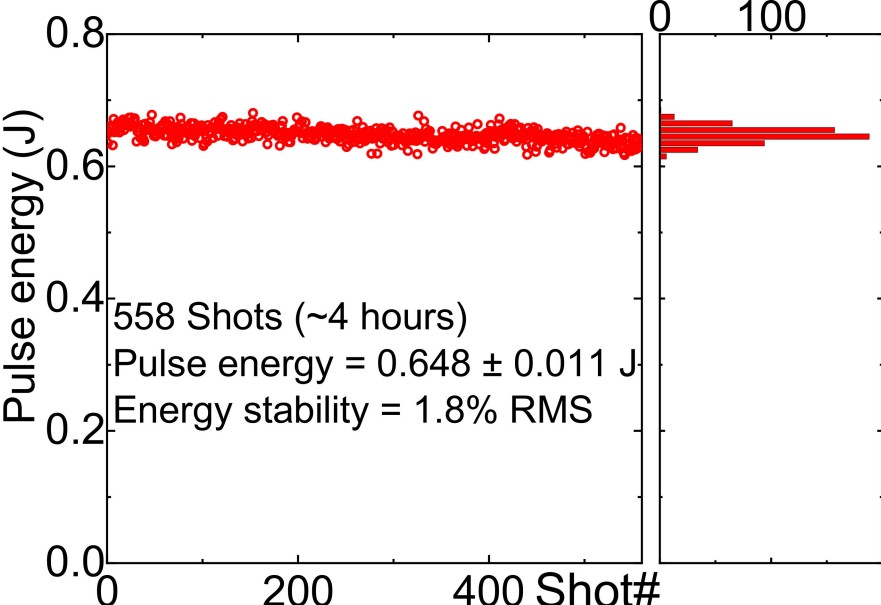

**Figure 6.** J-KAREN laser pulse energy stability during one day of an experiment (power amplifier mode, 4 h of operation). The pulse energy is calculated from the absolutely calibrated spectrum of the full-diameter beam.

Apart from the energy, rapidly-varying wavefront distortions due to air fluctuations, especially convection, in the long beamline also affect the shot-to-shot variations in the experiment. To suppress this, we have enclosed nearly the entire beamline in plastic boxes. The effect of the beamline covering is demonstrated in Figure 7, which shows histograms of the wavefront error of an 80-mm diameter Laser Diode (LD) beam for the cases of the open and enclosed beamline.

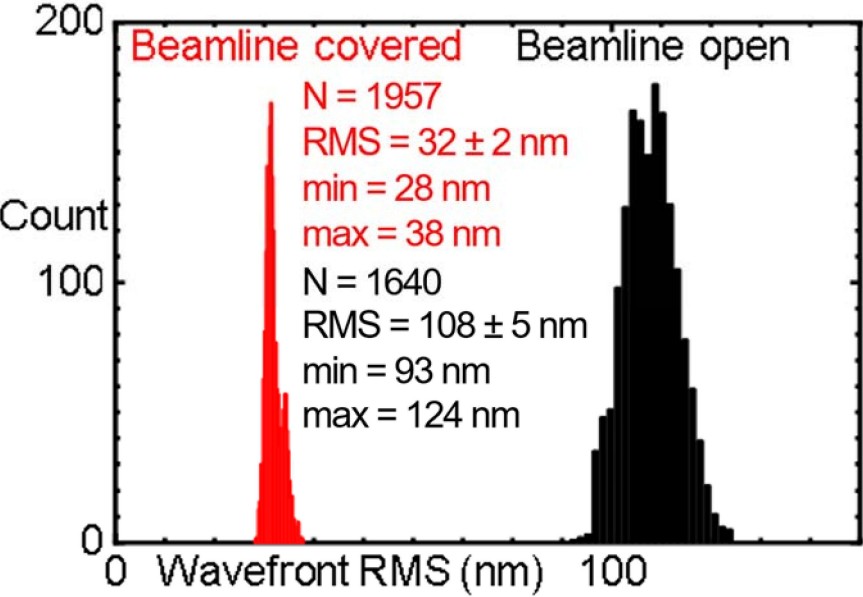

**Figure 7.** Effect of the beamline covering on the LD wavefront fluctuations.

We would like to emphasize the difference between the real (measured) focal spot and the one estimated from the measured wavefront, i.e., from the point-spread-function calculated from the wavefront. For example, for the 285-mm diameter alignment beam focused after the compressor with

an $f/7$ Off-Axis Parabolic mirror (OAP), the normalized irradiance estimated from the measured wavefront was 0.63 ± 0.06 (2000 frames average ± standard deviation); see Figure 8 (right red distribution). A typical PSF calculated from the wavefront data is shown in Figure 8, right inset. On the other hand, the real focal spot is shown in Figure 8 (left inset). The spot had a full width at half maximum (FWHM) size of just ~10% wider than the diffraction limit and a full width at $1/e^2$ level just ~6% wider. Despite this, the normalized irradiance estimated from the measured spot was 0.48 ± 0.03 (Figure 8, left blue distribution), i.e., notably lower than the PSF predicted from the wavefront data. This was due to: (i) high-frequency wavefront distortions, which were missed by the wavefront sensor, but produced a low-irradiance halo containing up to 50% energy around the main spot; and (ii) the absence of the wavefront data at the low-intensity beam edges, which is a known drawback of even state-of-the-art wavefront sensors [12]. An even larger difference can be seen in the case of a broadband pulse with spatiotemporal couplings, which are typically not detected by the wavefront sensors.

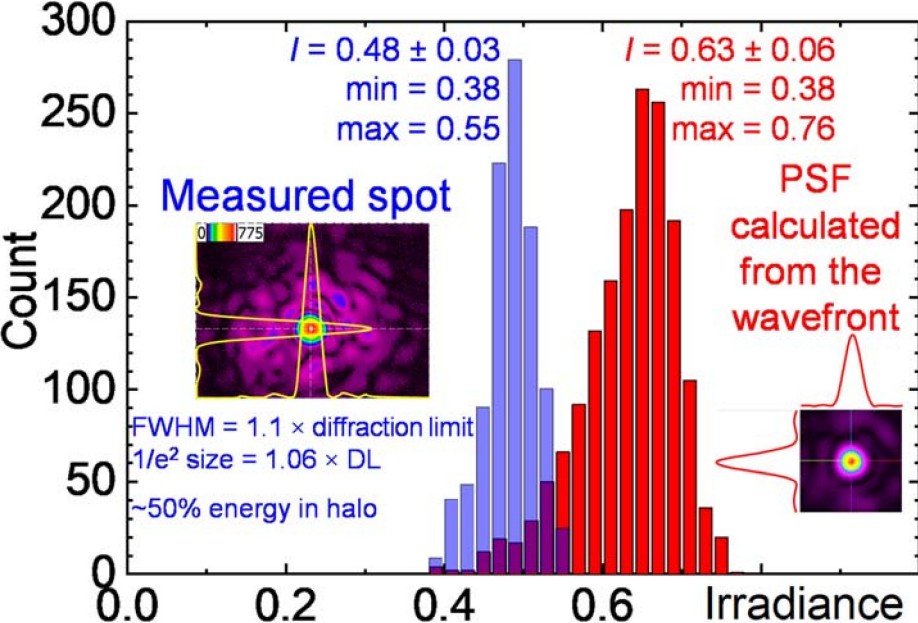

**Figure 8.** Normalized irradiance distributions obtained from the direct spot measurement (blue) and from Point-Spread Function (PSF) calculated from the measured wavefront (red). Examples of the measured spot and the calculated PSF are shown in the left and right insets, respectively. DL: diffraction limit.

## 4. Discussion

As we see from Figure 3, the spectrum (a) was significantly fainter than the spectra obtained in other campaigns. We suggest that this can be explained by laser filamentation, as the focal spot in this campaign (Figure 2a) was vertically elongated. This suggestion is supported by the vertical splitting of the X-ray spectra observed in some shots of this campaign. An example is shown in Figure 9a, where a relatively weak spectrum was split into three beamlets. Only one of the beamlets had a spectrum visible up to the photon energy of ~200 eV, while the other two beamlets were even weaker, and correspondingly, their spectra were visible above the noise up to ~160 eV only. The filamentation hypothesis is also supported by the shapes of the transmitted laser pulse footprints, Figure 10, which showed several laser beamlets. We note that the data in Figure 10a were obtained in the same shot with data in Figure 9a, and in the middle of the screen, there was an aperture for the X-ray spectrograph; also, the laser footprint was clipped from the sides. The data obtained during the same campaign with the setup modified in order to see the whole footprint are shown in Figure 10b.

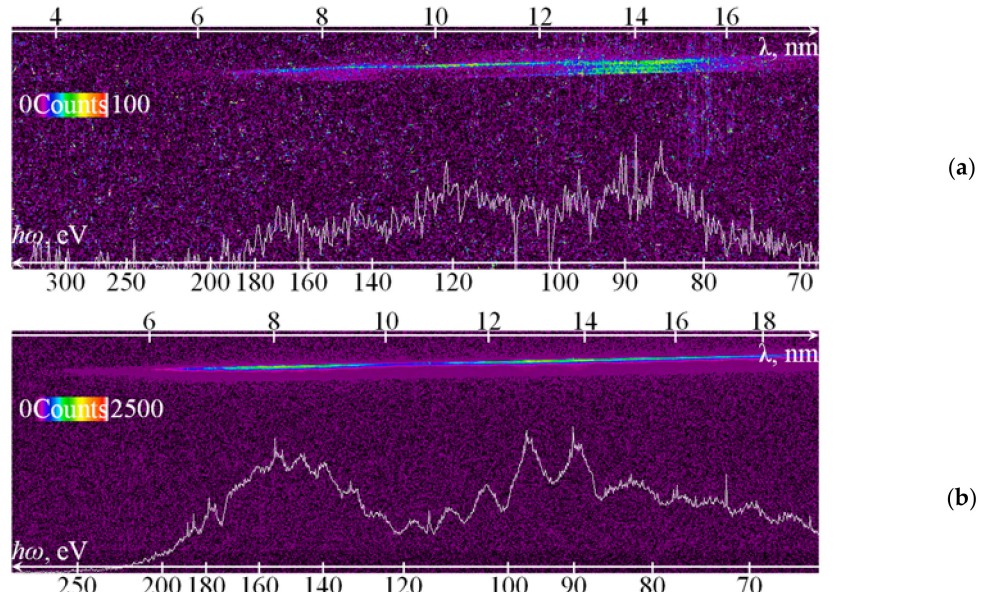

**Figure 9.** Examples of the raw data. (**a**,**b**) Data obtained in the experimental campaigns with the focal spots shown in Figure 2a,b, respectively.

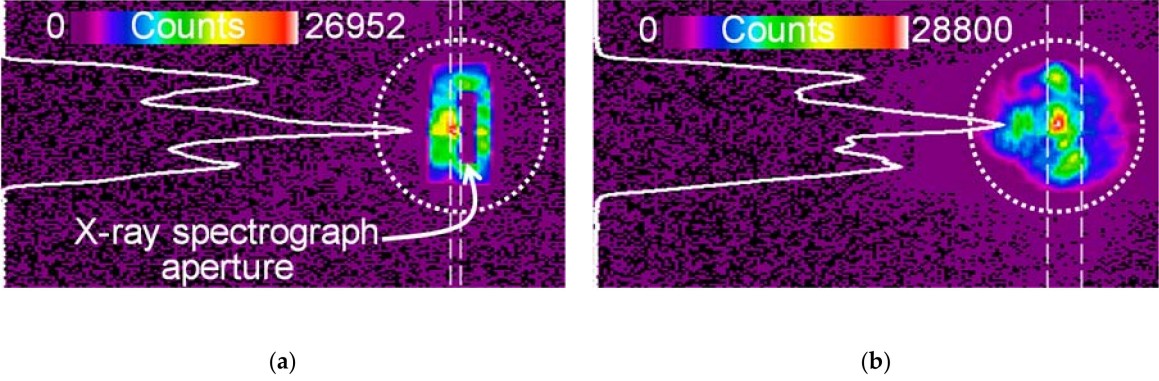

(**a**)    (**b**)

**Figure 10.** Transmitted footprints of the laser pulse on the Teflon screen situated at a distance of 63 cm after the gas jet (see Figure 1). The vertical lineouts on the left of the frames are drawn through the regions denoted with the dashed lines. The dotted circles have a diameter of 100 mm, i.e., an angular diameter of 9.1°. (**a**) The data taken simultaneously (the same shot) with the X-ray spectrum shown in Figure 9a; the dark rectangular area in the middle of the footprint corresponds to the aperture made in the screen to let the X-rays pass to the spectrograph; the footprint is clipped to the rectangular shape by other devices in the chamber. (**b**) The data obtained in the same experimental campaign as (**a**), but on another day, when the setup was modified to see the whole transmitted laser pulse footprint.

In other experimental campaigns, with more symmetric focal spots, the splitting of the X-ray spectra was not observed; typical data are shown in Figure 9b.

## 5. Conclusions

We presented the dependence of coherent X-ray generation by relativistic plasma singularities, via the BISER mechanism, on the laser irradiance. We found that the harmonic yield increased dramatically when the irradiance increased from ~$2 \times 10^{18}$ up to ~$10^{19}$ W/cm$^2$, which suggested that higher irradiances may be even more favourable. It is highly desirable to test this hypothesis experimentally.

We derived the requirements on the laser wavefront distortion and compressor alignment accuracy for the experiments on BISER. We described preparation of the recently upgraded J-KAREN-P laser for the experiments and reported on the present status of its stability and beamline quality.

**Acknowledgments:** We acknowledge financial support from the Japan Society for the Promotion of Science (JSPS) KAKENHI JP 20244065, JP 21740302, JP 23740413, JP 25287103, JP 25390135 and JP 26707031, the Ministry of Education, Culture, Sports, Science and Technology (MEXT), and the State Assignment of FASO of Russia to JIHT RAS (topic #01201357846). S.V. Bulanov acknowledges support from the project High Field Initiative (CZ.02.1.01/0.0/0.0/15 003/0000449) from the European Regional Development Fund.

**Author Contributions:** A.S.P. conceived of the research. The experiments were planned and prepared by A.S.P., M.K., H.D., A.S., K.O., E.N.R., D.N., T.I., M.N., N.H., T.K., K.K., Y.K. and S.V.B., led by A.S.P. and M.K. and performed by T.A.P., A.Ya.F., K.O., A.S., Y.H., H. Ko., D.N., H. Ki. and Y.F. The experimental data were analysed and interpreted by A.S.P., T.Zh. E., J.K.K., T.K., K.K. and M.K. A.S.P. led the manuscript writing. All authors contributed to the final version of the manuscript.

**Conflicts of Interest:** The authors declare no conflict of interest. The funding sponsors had no role in the design of the study; in the collection, analyses or interpretation of data; in the writing of the manuscript; nor in the decision to publish the results.

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
