# Peer review of "Laser Requirements for High-Order Harmonic Generation by Relativistic Plasma Singularities"

_qubs, doi:10.3390/qubs2010007_

Round 1

Reviewer 1 Report

Pirozhkov et al present a follow-up study to the landmark experiment published 5 years ago in Ref. 1, in which a new plasma-based mechanism - Burst Intensification by Singularity Emitting Radiation (BISER) for high harmonic generation in gas jets was reported. 

In the present work they examine ways of optimising the high harmonic emission into the important water window (282-533 eV), which offers optimal contrast for biological samples. They appear to concentrate chiefly on the laser irradiance, which for these short pulses is sensitive to the precise wavefront shape and angular chirp. The authors set limits on the tolerable focal spot size and reproducibility using the J-KAREN-P laser, providing useful guidance to users of comparable laser systems.

The experimental campaign also reveals anomalies in the spectrographs which the authors attribute to laser filamentation triggered by an elongated focal spot, pointing to structure in the transmitted light and multiple traces in the spectrogaph data. This remains speculative however, and would really require interferograms of the gas jet to confirm this explanation. Alternatively, might this not show up in PIC simulations with non-symmetric focal spots?

Author Response

Reply to the Reviewer 1 report:

Pirozhkov et al present a follow-up study to the landmark experiment published 5 years ago in Ref. 1, in which a new plasma-based mechanism - Burst Intensification by Singularity Emitting Radiation (BISER) for high harmonic generation in gas jets was reported.

In the present work they examine ways of optimising the high harmonic emission into the important water window (282-533 eV), which offers optimal contrast for biological samples. They appear to concentrate chiefly on the laser irradiance, which for these short pulses is sensitive to the precise wavefront shape and angular chirp. The authors set limits on the tolerable focal spot size and reproducibility using the J-KAREN-P laser, providing useful guidance to users of comparable laser systems.

We thank the Reviewer 1 for finding our results important and useful.

The experimental campaign also reveals anomalies in the spectrographs which the authors attribute to laser filamentation triggered by an elongated focal spot, pointing to structure in the transmitted light and multiple traces in the spectrogaph data. This remains speculative however, and would really require interferograms of the gas jet to confirm this explanation. Alternatively, might this not show up in PIC simulations with non-symmetric focal spots?

We agree with the Reviewer 1 that our suggestion that the data obtained with the poor spot quality can be attributed to filamentation remains speculative. We made this clear by writing in the Discussion section "We suggest that this can be explained by laser filamentation,..." and "The filamentation hypothesis is also supported by the shapes of the transmitted laser pulse footprints..." We also agree that a rigorous proof of this hypothesis would require high-resolution interferograms and/or PIC simulations with realistic non-ideal spots. However, at present these are not available, so we limit ourselves to the brief discussion of this hypothesis in the Discussion section. We note that our manuscript title is "Laser Requirements for High-Order Harmonic Generation by Relativistic Plasma Singularities," and our main goals are to "examine ways of optimising the high harmonic emission," and to "set limits on the tolerable focal spot size and reproducibility," as it is correctly formulated by the Reviewer 1 in the previous paragraph. We therefore concentrate on our main goals and leave the proof of the filamentation hypothesis to future publications devoted to plasma physics and laser-plasma interaction.

Reviewer 2 Report

The manuscript is concerned with laser parameter optimization which is of crucial importance for laser plasma experimental science. The main focus of the investigation is shifted to the focal spot optimization as well as the elimination of the angular chirp. The perfection of the laser pulse generated by the J-KAREN laser system is tested with respect to efficiency of BISER x-ray source.

The presented technical details might be interesting for a number of laser-plasma experimentalists. But, in my opinion, there are some principle points to be addressed before the paper can be published.

1. The authors avoid in the manuscript the description of the gas jet density profile. In addition, no explanation is given on the mechanism of X-Ray generation. The accepted Sci. Rep paper [6] will hopefully shed light on the details, but there is almost nothing to rely on now. Why is BISER dominant over betatron under these particular experimental conditions? For example, in the paper of J. Wenz [NATURE COMMUNICATIONS | 6:7568 | DOI: 10.1038/ncomms8568], the parameters of the laser pulse and the plasma density are very close to those presented, while the mechanism of X-Ray generation is completely different.

2. In the work, the authors compare the spectra from two series of experiments (a) and (b). The campaigns differ in both the focal spot morphology and the peak irradiance (and possibly the point in the gas jet is different as well as plasma density), while the authors conclude that the main role is played by irradiance. In this case, the differences in the form of the focal spots and some other factors (such as the position of the gas jet, and the contrast of the laser pulse) are supposed to be insignificant, which is not justified.

There are a few other issues the authors should consider:

1) lines 64-65: The terms “angular dispersion” and “angular chirp” have to be clarified at the first appearance

2)         Figure 3: Space should be added between “300” and 'hω”

3)         The presented requirements for the waterfront distortions are in good consistency with well-known Marshal criterion. I would recommend underlining this.

4)         Line 167. “Ti:Sapphire beam” term should be clarified

5)         Fig 8. The text is overlapped with the lines. The face is too sad.

6)         The presence of Figure 10 (a) raises more questions than gives answers.

7)         The internet links for refs. 2, 7, 9, 12 and 14 are not valid.

Author Response

Reply to the Reviewer 2 report:

The manuscript is concerned with laser parameter optimization which is of crucial importance for laser plasma experimental science. The main focus of the investigation is shifted to the focal spot optimization as well as the elimination of the angular chirp. The perfection of the laser pulse generated by the J-KAREN laser system is tested with respect to efficiency of BISER x-ray source.

The presented technical details might be interesting for a number of laser-plasma experimentalists. But, in my opinion, there are some principle points to be addressed before the paper can be published.

We thank the Reviewer 2 for acknowledging that our manuscript might be interesting for experimentalists.

1. The authors avoid in the manuscript the description of the gas jet density profile. In addition, no explanation is given on the mechanism of X-Ray generation. The accepted Sci. Rep paper [6] will hopefully shed light on the details, but there is almost nothing to rely on now. Why is BISER dominant over betatron under these particular experimental conditions? For example, in the paper of J. Wenz [NATURE COMMUNICATIONS | 6:7568 | DOI: 10.1038/ncomms8568], the parameters of the laser pulse and the plasma density are very close to those presented, while the mechanism of X-Ray generation is completely different.

The gas jet density profile has been described in our previous papers. We prefer to avoid duplication of figures and therefore refer to [2] where this information has been published; we note that [2] is an open access paper. We added the following sentence to the revised manuscript (page 2, lines 80-81 of the revised manuscript with the tracked changes): "typical gas jet density profiles are shown in Fig. 2 of Ref. [2]."

We added the following description requested by the Reviewer 2 and Reviewer 3 (page 2, lines 57-62 revised manuscript): "This emission originates from extremely localized, nanometer-scale, regions where the local electron density is several orders of magnitude higher than the original plasma density. This results in constructive interference, i.e. x-ray yield proportional to the number of electrons squared. The large number of photons together with the small source size and attosecond duration provide very high brightness, e.g. 1027 photons/mm2∙mrad2s in 0.1% bandwidth at the wavelength of 18 nm [6]." Also, our Scientific Reports paper [6] describing the details of the BISER mechanism is now published and is also free to download.

The Reviewer 2 is right in that there exist several x-ray generation mechanisms, such as betatron, Bremsstrahlung, emission of plasma ions, etc. In order to clarify our conditions for the readers, we added the following sentences in the revised manuscript (page 3, lines 101-105 revised manuscript): "In a wide region of parameter space around these optimum conditions, the BISER was a dominant mechanism of x-ray generation in the spectral region of interest. We note that our plasma length was kept relatively short, typically ~1-2 mm, which was significantly shorter than in experiments for electron acceleration and betatron x-ray generation with similar lasers [Wenz et al Nature Comm. 2015, 6, 7568]."

2. In the work, the authors compare the spectra from two series of experiments (a) and (b). The campaigns differ in both the focal spot morphology and the peak irradiance (and possibly the point in the gas jet is different as well as plasma density), while the authors conclude that the main role is played by irradiance. In this case, the differences in the form of the focal spots and some other factors (such as the position of the gas jet, and the contrast of the laser pulse) are supposed to be insignificant, which is not justified.

The Reviewer 2 is right that several factors affect the BISER, including the spot shape, peak irradiance, focusing conditions, and plasma density, as correctly mentioned in his/her comment. This dependence will be reported in our future publications devoted to the details of the BISER mechanism. Here our attention is focused on the following question: what is the maximum x-ray yield, and how does it depend on spot and irradiance. As for the contrast, its influence is not studied systematically and it is too early to make comments about it. We added the following sentences to the revised manuscript (page 3, lines 98-100 revised manuscript): "Harmonic generation by the BISER mechanism depended on several factors, most importantly on the plasma density and gas jet position with respect to the laser focus. The details of these dependences will be reported elsewhere."

There are a few other issues the authors should consider:

1) lines 64-65: The terms “angular dispersion” and “angular chirp” have to be clarified at the first appearance

We added the requested explanation (page 2, lines 69-72 revised manuscript) as follows: "The angular dispersion, i.e. the dependence of propagation direction on wavelength, should be minimized, so that the difference between propagation angles should be kept significantly smaller than the diffraction divergence, i.e. μrad level for 100 to 300 mm beam diameters. The corresponding angular chirp, i.e. the derivative of the propagation angle on wavelength, should not exceed ~10-2 μrad/nm for 40 nm bandwidth."

2)         Figure 3: Space should be added between “300” and 'hω”

We moved the axis label down and replaced "ћω" with "Photon energy" to clarify the figure.

3)         The presented requirements for the waterfront distortions are in good consistency with well-known Marshal criterion. I would recommend underlining this.

We made the requested comparison on page 5 (lines 149-150 revised manuscript): "...the wavefront distortion should be kept smaller than about 100 nm rms, i.e. smaller than λ/8, which is close to the Maréchal criterion of a diffraction-limited beam."

4)         Line 167. “Ti:Sapphire beam” term should be clarified

We changed it to "J-KAREN-P beam" to clarify the meaning, as requested by the Reviewer 2 (page 5, line 184 revised manuscript).

5)         Fig 8. The text is overlapped with the lines. The face is too sad.

We revised the figure so that the text is now not overlapped. Unfortunately, there is nothing we can do with the sad face.

6)         The presence of Figure 10 (a) raises more questions than gives answers.

We changed the arrow in Figure 10 (a) and clarified the caption as follows: "Transmitted footprints of the laser pulse on the Teflon screen situated at the distance of 63 cm after the gas jet, Figure 1. The vertical lineouts on the left of the frames are drawn through the regions denoted with the dashed lines. The dotted circles have the diameter of 100 mm, i.e. the angular diameter of 9.1°. (a) The data taken simultaneously (the same shot) with the x-ray spectrum shown in Figure 9 (a); the dark rectangular area in the middle of the footprint corresponds to the aperture made in the screen to let the x-rays pass to the spectrograph; the footprint is clipped to the rectangular shape by other devices in the chamber."

We also added "the x-rays passed to the spectrograph through the aperture made in the screen" in Figure 1 caption.

7)         The internet links for refs. 2, 7, 9, 12 and 14 are not valid.

We thank the Reviewer 2 for noticing the invalid links. We have updated these links in the revised version.

Reviewer 3 Report

The topic of the MS indeed touches a very important aspect, the high harmonics generation from gas jet targets with relativistic laser pulses. First of all, it shows that the harmonic generation increased with the laser irradiance. Secondly, they preformed experiments examining two important parameters, the role of rms wavefront error and angular dispersion, and also shown the experimental realization from J-KAREN-P laser. This work is very interesting and will certainly be of interest to the community. However, there are a number of questions stays unclear, the authors should clarify them before the MS being considered to be published.

1)   In the introduction, the authors should briefly explain the BISER mechanism.

2)   It states that “several experimental campaigns with the J-KAREN laser”, while the details of laser parameters are not shown in the MS. Please give some details in describing the laser system, for example, the energy, pulse duration, wavelength, etc.

3)   Fig. 3, the measured x-ray spectra was shown up to 350 eV, is that the detection limit for the spectrometer? Also, in the y axis, what is the unit? eV? please delete the hω.

4)   One major result of this work is the observation that the high harmonics generation increased with the laser irradiance. While the discussion of the underlying physcis leading to the enhancement seems to be lacking, the authors should explain it in more details.

5)   The influence of rms wavefront error is shown in Fig. 4, it is also interesting that there are more harmonics yields under the ‘astigmatism’ situation. Could the authors comment on that briefly? Why does it occur? Does it mean that the ‘out-of-focus’ is better for BISER, which might significantly reduce the requirement for the laser intensity?

6)   In the MS, several beamlets were observed in the harmonics spectrum and are attributed to the laser pulse profile, as shown in Fig. 9 and Fig. 10. Fig. 9 show that there is one strong beamlet which has a spectrum up to 200 eV, the other two are located below the stronger one, with weak signals up to 160 eV. Fig. 10 is the footprint of the transmitted laser pulse, which also contain 3 beamlets. While the middle one is the most intense one, with two weak ones in each side. One question here, why the location of the laser beamlets deviated from the harmonic beamlets? I could image that the strongest laser peak gave the highest harmonic yield.

Author Response

Reply to the Reviewer 3 report:

The topic of the MS indeed touches a very important aspect, the high harmonics generation from gas jet targets with relativistic laser pulses. First of all, it shows that the harmonic generation increased with the laser irradiance. Secondly, they preformed experiments examining two important parameters, the role of rms wavefront error and angular dispersion, and also shown the experimental realization from J-KAREN-P laser. This work is very interesting and will certainly be of interest to the community. However, there are a number of questions stays unclear, the authors should clarify them before the MS being considered to be published.

We thank the Reviewer 3 for finding our subject important and our manuscript very interesting.

1)   In the introduction, the authors should briefly explain the BISER mechanism.

Following the Reviewer 2 and Reviewer 3 requests, we added the following description in the introduction (page 2, lines 57-62 revised manuscript): "This emission originates from extremely localized, nanometer-scale, regions where the local electron density is several orders of magnitude higher than the original plasma density. This results in constructive interference, i.e. x-ray yield proportional to the number of electrons squared. The large number of photons together with the small source size and attosecond duration provide very high brightness, e.g. 1027 photons/mm2∙mrad2s in 0.1% bandwidth at the wavelength of 18 nm [6]." Also, our Scientific Reports paper [6] describing the details of the BISER mechanism is now published and is free to download.

2)   It states that “several experimental campaigns with the J-KAREN laser”, while the details of laser parameters are not shown in the MS. Please give some details in describing the laser system, for example, the energy, pulse duration, wavelength, etc.

We expanded the laser description as follows (page 2, lines 78-79 revised manuscript): "The laser wavelength was ~0.8 μm, the pulse energy varied from 0.4 to 0.8 J, the effective pulse duration from 34 to 160 fs, and the peak power from 5 to 20 TW."

3)   Fig. 3, the measured x-ray spectra was shown up to 350 eV, is that the detection limit for the spectrometer? Also, in the y axis, what is the unit? eV? please delete the hω.

The detection limit of the spectrograph depended on the x-ray yield. Indeed, this limit, determined by the noise level, was ~350 eV for the spectrum [1]. However, it was ~210 eV for the spectrum (a) due to the weaker signal. The spectrum (b) actually continued till ~500 eV, which was the hard edge of the spectrograph range determined by the CCD width. The unit of the vertical axis is "nJ/(eV sr)", as shown at the top of the axis. We moved the horizontal axis label down and replaced "ћω" with "Photon energy" to clarify the figure.

4)   One major result of this work is the observation that the high harmonics generation increased with the laser irradiance. While the discussion of the underlying physcis leading to the enhancement seems to be lacking, the authors should explain it in more details.

We note that our manuscript title is "Laser Requirements for High-Order Harmonic Generation by Relativistic Plasma Singularities," and our main goals are to report the dependence of the maximum achievable x-ray yield on the irradiance, and to derive the requirements on the laser system for efficient coherent x-ray generation via the BISER mechanism. We therefore concentrate on our main goals and leave the description of the underlying physics, including laser propagation, self-focusing, singularity formation, and BISER emission in presence of aberrations and spatiotemporal couplings, to future publications.

5)   The influence of rms wavefront error is shown in Fig. 4, it is also interesting that there are more harmonics yields under the ‘astigmatism’ situation. Could the authors comment on that briefly? Why does it occur? Does it mean that the ‘out-of-focus’ is better for BISER, which might significantly reduce the requirement for the laser intensity?

Figure 4 shows the measured dependence of the irradiance on the rms wavefront distortion, not the dependence of the BISER yield on it; we are sorry for the confusion. As Reviewer 3 has correctly noticed, from that figure it may actually follow that an astigmatic laser beam could be better than a beam with higher-order aberrations with the same rms distortion. It would be indeed interesting to test this experimentally. However, up to now we have not tried this in our experiments. To avoid confusion of the readers, we deleted this data set from Figure 4 and the corresponding sentence from the text. Our conclusions are not affected by this deletion.

6)   In the MS, several beamlets were observed in the harmonics spectrum and are attributed to the laser pulse profile, as shown in Fig. 9 and Fig. 10. Fig. 9 show that there is one strong beamlet which has a spectrum up to 200 eV, the other two are located below the stronger one, with weak signals up to 160 eV. Fig. 10 is the footprint of the transmitted laser pulse, which also contain 3 beamlets. While the middle one is the most intense one, with two weak ones in each side. One question here, why the location of the laser beamlets deviated from the harmonic beamlets? I could image that the strongest laser peak gave the highest harmonic yield.

Indeed, it would be very interesting to know the details of the relation between the laser propagation and BISER emission. Reviewer 3 is right, our observations seem to suggest that the strongest strip in Figure 9 (a) probably originates from the side beamlet with relatively low energy. From this it may follow that the BISER mechanism is determined by singularity formation conditions and electron acceleration, rather than on the energy contained in the whole driving pulse. However, because the filamentation is our hypothesis, not a conclusion, we postpone the detailed discussion to our future publications.
